# The Case for Bisphosphonate Use in Astronauts Flying Long-Duration Missions

**DOI:** 10.3390/cells13161337

**Published:** 2024-08-13

**Authors:** Reece Rosenthal, Victor S. Schneider, Jeffrey A. Jones, Jean D. Sibonga

**Affiliations:** 1Center for Space Medicine, Baylor College of Medicine, Houston, TX 77030, USA; rmrosenthal@houstonmethodist.org (R.R.); vschneider@nasa.gov (V.S.S.); jeffrey.jones9@va.gov (J.A.J.); 2Space Operations Mission Directorate, Human Research Program, NASA Mary W. Jackson Headquarters, Washington, DC 20546, USA; 3Human Health & Performance Directorate, NASA Johnson Space Center, 2101 NASA Parkway SK3, Houston, TX 77058, USA

**Keywords:** microgravity, immobilization, weightlessness, disuse, osteoclasts, osteoblasts

## Abstract

Changes in the structure of bone can occur in space as an adaptive response to microgravity and on Earth due to the adaptive effects to exercise, to the aging of bone cells, or to prolonged disuse. Knowledge of cell-mediated bone remodeling on Earth informs our understanding of bone tissue changes in space and whether these skeletal changes might increase the risk for fractures or premature osteoporosis in astronauts. Comparisons of skeletal health between astronauts and aging humans, however, may be both informative and misleading. Astronauts are screened for a high level of physical fitness and health, are launched with high bone mineral densities, and perform exercise daily in space to combat skeletal atrophy as an adaptive response to reduced weight-bearing function, while the elderly display cellular and tissue pathology as a response to senescence and disuse. Current clinical testing for age-related bone change, applied to astronauts, may not be sufficient for fully understanding risks associated with rare and uniquely induced bone changes. This review aims to (i) highlight cellular analogies between spaceflight-induced and age-related bone loss, which could aid in predicting fractures, (ii) discuss why overreliance on terrestrial clinical approaches may miss potentially irreversible disruptions in trabecular bone microarchitecture induced by spaceflight, and (iii) detail how the cellular effects of the bisphosphonate class of drugs offer a prophylactic countermeasure for suppressing the elevated bone resorption characteristically observed during long-duration spaceflights. Thus the use of the bisphosphonate will help protect the bone from structural changes while in microgravity either along with exercise or alone when exercise is not performed, e.g. after an injury or illness.

## 1. Introduction

### 1.1. Bone Anatomy and Physiology

Long bones such as the femur are comprised of both cortical and trabecular bone. The tubular portion of long bones, the diaphysis, is predominantly made up of cortical bone, which forms a compact shell encasing the bone marrow compartment. Aside from protecting the cellular milieu of the bone marrow, cortical bone contributes to the ability of the whole bone to resist mechanical forces (e.g., bending and torsional forces, muscular forces). Trabecular bone, a highly porous bone located in the medullary canal at the ends of long bone and within the body of the vertebrae, provides a compressible, load-distributing function [1]. Its high content of bone surfaces is easily accessible by bone-resorbing osteoclasts in bone marrow, enabling trabecular bone to serve as a reservoir to maintain mineral homeostasis, releasing calcium and other minerals during the resorption of bone. 

### 1.2. Osteoporosis: Pathophysiology and Treatment

Primary osteoporosis is a condition of skeletal fragility resulting from greater resorption than formation of bone tissue during the normal remodeling of skeletal tissue in adult humans. Type 1 primary osteoporosis is associated with increased bone resorption by osteoclasts stimulated with the deficiency of estrogen [2] in post-menopausal women while Type 2 primary osteoporosis is reflective of declining production of mineralized bone matrix with increasing age [3,4,5]. While the effect of spaceflight on circulating estrogen has not been fully investigated in astronauts, sub-optimal nutrition and reduced physical activity are clear risk factors for skeletal deconditioning during spaceflight. Hence, the optimization of dietary intake and exercise performance remains the first line of countermeasure defense for an astronaut cohort selected for its pristine medical health and physical fitness. 

Primary osteoporosis is characterized by a proportionally greater trabecular bone loss compared to cortical bone loss [6]. The pronounced trabecular changes in part are due to the relatively increased vascularity and surface area of trabecular bone in the medullary canal, which is easily accessible to circulating, bone-resorbing osteoclasts. Skeletal fragility, a hallmark of osteoporosis, results from reductions in bone mass, increased porosity in cortical bone, and microarchitectural changes in trabecular bone [7]. Fractures due to fragility have been characteristically observed at skeletal sites with high trabecular bone content, e.g., the vertebrae, wrist, and hip [8].

Changes in bone mass at distinct skeletal sites can be quantified by using dual-energy X-ray absorptiometry (DXA), which measures areal bone mineral density (aBMD, g/cm^2^). A decline in aBMD is a key biomarker for skeletal fragility associated with primary osteoporosis, and increased fragility fractures are most evident in post-menopausal women and men older than 50 years of age. An aBMD T-score ≤ −2.5 for hip and spine sites is predictive of fragility fractures for this age group and serves as a cut-point for identifying persons-at-risk. Whether these same guidelines identify astronauts who may benefit clinically from prophylactic mitigation is not readily apparent, i.e., the aBMD T-score has not been associated with fragility fractures in the actively flying astronaut [9].

Persons at risk for primary osteoporosis can be treated with anti-resorptive pharmacotherapy, such as bisphosphonates, with the goal of providing prophylaxis against fracture. The cellular action of the bisphosphonate class of anti-resorptives is like naturally occurring cellular pyrophosphates. The negatively charged ligands of the bisphosphonate molecule impart varying affinities for the collagen matrix, which may likely influence the varying potencies of different bisphosphonate drugs. In the process of bone resorption, osteoclasts degrade bone matrix releasing the bisphosphonate molecule from bone where it is internalized by the osteoclast. Once integrated into the osteoclast, the bisphosphonate inhibits several intracellular enzymes critical to osteoclastic bone-resorbing activity [10,11]. By decreasing osteoclast activity and number, bisphosphonates can help restore the balance between bone resorption and bone formation within the bone remodeling unit for metabolic bone disorders that are characterized by elevated bone resorption (e.g., menopause-induced osteoporosis). At the tissue level, bisphosphonates suppress the loss of bone mass subsequently protecting skeletal tissue from structural changes, helping to preserve the overall integrity of bones and mitigating the increased risk of fracture associated with aging [12,13].

The bisphosphonate alendronate was approved by the Food and Drug Administration for preventing and treating osteoporosis in post-menopausal women in 1996 and in men in 2000 [14,15,16]. The adverse effects of the drug class are well known [17]. Serious side effects include osteonecrosis of the jaw (ONJ) and atypical femur fractures. ONJ risk is higher in those who need dental work or in patients on long-term intravenous bisphosphonate therapy for preventing skeletal metastases from cancers. Therefore, bisphosphonates are recommended to be given after dental work has been completed [18]. In the osteoporosis patient population on oral bisphosphonates, the prevalence of ONJ is 0.01–0.04% [19]. In a registry-based cohort study with 9956 subjects at risk for or diagnosed with osteoporosis, the incidence rate of atypical femur fractures was considered low (7.9/10,000 person-years) with bisphosphonate treatment and comparable to the 7.1/10,000 person-years in patients treated with denosumab, a rank-ligand inhibitor [20]. However, oral bisphosphonates demonstrate poor oral bioavailability, possible gastrointestinal irritation, and non-specific absorption to compounds in the diet, making them a less attractive candidate for use in austere space environments where food options are minimized [21]. As an alternative, an intravenous bisphosphonate would guarantee adequate dosing with long-lasting anti-resorptive activity [22].

## 2. The Space Environment

### 2.1. Effects of Microgravity on Bone

The atrophy of bone during long-duration spaceflight causes a loss of bone mass that could contribute to skeletal fragility [23]. Prior to the availability of resistive exercise (up to 600 lb-f) during spaceflight, large deficits in DXA-measured BMD were observed postflight although the clinical significance in younger aged men and women was not yet defined [24,25]. As much as a 26% loss in BMD was observed in the hip during long-duration spaceflight [Personal Communication, V Schneider 2022] [26]. This loss was most prominent at the lower spine and hip, which logically follows as these structures are no longer bearing weight in microgravity (with Wolff’s law, “form follows function” being applicable in this case) [27]. Mechanoreceptors in the bone play an important signaling role and regulate the balance of bone homeostasis on Earth. In space, skeletal forces are minimal, and the balance is tipped towards bone atrophy, leading to skeletal deconditioning and, if unabated, to skeletal fragility. In addition to bone’s adaptation to its reduced weight-bearing function, there are both putative (e.g., increased atmospheric CO_2_ and possible hypercapnia) and established risk factors for bone loss (e.g., reduced calcium intake) that may be exacerbated during extended spaceflight sojourns beyond low Earth orbit [21,28,29].

### 2.2. Countermeasures to Mitigate Spaceflight-Induced Osseous Changes: Resistive Exercise

As a result of the early research demonstrating aBMD loss during spaceflight, preflight and postflight DXA measurements have been required in astronauts flying long-duration spaceflights on the International Space Station (ISS) to assess for the early onset of osteoporosis [25]. Additionally, astronauts have been required to perform vigorous resistive exercise throughout the duration of the mission to serve as prophylaxis against deficits in postflight bone mineral densities. The ISS’s Advanced Resistive Exercise Device (ARED) which provides resistive loads in space, is roughly equivalent to heavy weight-bearing exercise on Earth [30]. As seen in Figure 1, improved resistance training with ARED exercise attenuates previously observed declines in DXA-measured bone mineral density after the usual 6-month missions [31]. Additionally, postflight T-scores <−2.5 ((the diagnostic criterion for age-related skeletal fragility) were also not evident in ISS astronauts [9,31] supporting the perception that ISS astronauts do not require additional mitigation beyond resistive exercise.

The actual impact of resistive exercise on the cellular mediation of bone remodeling is a relative unknown without bone biopsies to histologically assess changes occurring in-flight. The cellular response of bone to a reduced weight-bearing function has been characterized in iliac crest bone biopsies of subjects skeletally unloaded by 12 weeks of bed rest [34] where indices of bone resorption (e.g., osteoclast-covered surfaces and eroded surfaces in both trabecular and cortical bone) were increased while the osteoblast-covered surfaces were reduced [34]. As a surrogate for histological testing, biochemical markers of bone turnover—assayed in astronaut urine and blood samples acquired preflight/inflight/postflight—have collectively supported that the loss of bone mineral is the result of unbalanced and uncoupled cellular processes for resorption (stimulated) and formation (suppressed) during bone remodeling, contributing to a net loss of bone mass [31,35,36].

Even with the maintenance of preflight skeletal bone mass with resistance exercise, there is still an increased excretion of resorption biomarkers—N-telopeptide and helical peptide—suggesting an inability of resistive exercise to suppress the elevation of osteoclast-mediated bone resorption observed during spaceflight [36] and depicted in Figure 2 [35]. Conversely, astronauts using ARED display increases in serum levels of proteins and peptides, such as the osteoblast-specific bone alkaline phosphatase (BAP) (Figure 3) [35,36] reflecting either an increase in, or activation of, osteoblast number or an increase in osteoblast bone-forming activity. Therefore, these biochemistry data support the inference that resistive exercise stimulates bone formation but does not have a suppressive effect on the stimulated bone resorption observed during spaceflight [35].

### 2.3. Other Measures of Osseous Morphology Demonstrate Deficiencies in ARED

While DXA-observed BMD may appear to be preserved during spaceflight, pre- and postflight DXA measurements are static and do not capture the dynamics of in-space changes in bone mass. DXA averages bone mineral densities (e.g., grams mineral/cm^2^ projected bone area) across both cortical and trabecular bone compartments. Astronaut data from multiple flight studies using QCT indicate that cortical bone has a volumetric bone mineral density (vBMD) that is approximately 4-fold greater than the vBMD of trabecular bone in the proximal femur (personal communication of unpublished data, Sibonga, 2022). Over the last two decades, the DXA-based changes observed in astronauts have been more closely defined with QCT, which quantifies spatial tissue volume and density (vBMD, g/cm^3^) allowing for distinct assessments of trabecular and cortical bone compartments. Reductions in bone quality (e.g., loss and delayed/absence of recovered hip trabecular bone mass) have been observed [24,35,37,38,39,40,41]. Structural changes due to microgravity have been suggested, with more profound loss of trabecular bone compared to cortical bone (Figure 1) which was first reported in cosmonauts [37]. In this early study of Russian cosmonauts, cortical and trabecular volumetric BMD [vBMD] of the hip was measured at pre-flight, post-flight, and 6-month post-flight recovery period in crewmembers who had spent > 6 months in space. It was found that not only did the cosmonauts lose hip bone density while in space, but also that there was no significant recovery in BMD 6 months post-flight [37]. A subsequent study in astronauts serving 4–6-month missions used a similar methodology to assess 1-year post-flight recovery of hip vBMD. While there was significant recovery of total bone mass (perhaps contributing to an apparent unchanged DXA BMD), there was incomplete recovery of trabecular vBMD and estimated bone strength [41].

The longest follow-up study of note in long-duration astronauts that assessed measures of bone strength was after 2–4.5 years of recovery time on Earth and found that some measured attributes of whole bone strength did recover. Femoral trabecular bone density did not recover completely over the study period and further losses in density occurred 1–3.5 years after return to Earth—a finding which suggests that the physiology of bone remodeling may be profoundly altered by spaceflight in unpredictable ways [38].

QCT further documented that trabecular mass does not return to pre-flight status after 2 years back on Earth in 4 of the 10 astronauts who participated in a pilot demonstration of QCT for surveillance of full recovery [33]. Changes documented by QCT scans have suggested an increased risk for bone fracture through the analysis of finite element models generated from QCT data [42]. It has also been postulated that QCT, because of its detection of distinct changes in bone mass in three dimensions in both cortical and trabecular bone, would be a more accurate predictor of fracture risk compared to DXA [43,44,45,46] To summarize, in contrast to previous reports of postflight recovery of DXA-measured aBMD [26], distinct changes in trabecular bone, associated with accelerated loss of aBMD [24,33,47], could lead to irreversible degradation of trabecular microarchitecture. If DXA were the sole test to evaluate bone loss, bone recovery, and the efficacy of countermeasures then loss and recovery to trabecular bone would not be detected or monitored. 

The adaptation of bone structure and mass to spaceflight conditions therefore does not affect all bone compartments equally [32,33,41]. There are differences in cellular mediation of bone turnover between skeletal regions—which are likely influenced by the degree of weight-bearing function between Earth and space for specific bones or by the percentage content of trabecular or cortical bone within the whole bone which is discernable by QCT densitometry [25,48]. To date, we have observed that exercise is unable to suppress the spaceflight-induced elevation of bone resorption biomarkers [35]. Further analysis of QCT has shown that resistive exercise on ARED alone does not suppress the measurable loss of trabecular bone mineral density and as such may not be able to address underlying microarchitectural changes in trabecular bone which could contribute to fragility (Figure 1, representing the totality of all published NASA QCT data) [24,32,33,35]. Mechanical loading by ARED may increase total bone mass (detected by DXA) by activating osteoblasts on the periosteal surface of the cortex, not unlike the effect of re-ambulation on Earth after spaceflight [41]. The specific impact of ARED loading on trabecular bone is not detectable by DXA and may not be beneficial to this highly porous bony tissue [33,38,41]. To summarize, ARED may offset total bone loss during spaceflight by tilting the balance of bone turnover back in favor of bone formation, however, the use of DXA-measure BMD alone cannot detect detriments to sub-regional bone—specifically trabecular bone mass and trabecular microarchitecture. The delayed recovery and continued compartmental bone loss observed in the trabecular bone of the spine and hip after return to Earth [33,49] suggest that monitoring the effects of in-flight countermeasures in trabecular bone is warranted.

### 2.4. Countermeasures to Mitigate Spaceflight-Induced Osseous Changes: Bisphosphonates

Bisphosphonates are considered a first-line pharmacologic candidate for testing under spaceflight conditions given their successful suppression of bone loss and prevention of terrestrial complications of osteoporosis. Additionally, because the cellular mechanism of bisphosphonates involves uptake by the osteoclast, the cells that resorb and degrade bone tissue, it stands to reason that bisphosphonates would be able to address some of the deficiencies of ARED. Mechanical loading to skeletal bones by resistive exercise on ARED does not suppress elevated bone resorption during spaceflight, which has been consistently reported by biochemical data of bone turnover obtained from urine and blood specimens collected from astronauts during spaceflight [33,35,36].

Spaceflight studies have characterized the effects of bisphosphonates in the ISS astronaut population when used in conjunction with ARED. Multiple flight studies enabled comparison of ARED exercise relative to ARED exercise supplemented with the bisphosphonate alendronate (ARED + BP) during long-duration spaceflights. The combined therapies (ARED + BP) were associated with significant attenuation or complete protection in postflight aBMD of crewmembers compared with those who used only ARED alone. Studies utilizing QCT collectively demonstrate that ARED + BP significantly attenuates postflight trabecular vBMD deficit in total hip, trochanter, and femoral neck measurements [24,32,35] (Figure 1).

Biochemically, in contrast to those who used solely ARED or the interim Resistive Exercise Device (iRED also denoted as Pre-ARED), crewmembers who participated as treated subjects in the bisphosphonate flight study, were shown to excrete reduced levels of bone resorption biomarkers suggesting that attenuated deficits in postflight BMDs are due to a direct drug effect on the resorbing activity or number of osteoclast cells (Figure 2 and Figure 3) [35]. This reduction in the cellular breakdown of bone tissue might also preserve the terrestrial bone structure while in space, evidence that could be further supported by sensitive densitometry by QCT (Figure 1) and by high-resolution peripheral QCT [40].

## 3. Consequences and Implications for Decisionmakers

Most people who terrestrially experience osseous fragility are those at risk for primary osteoporosis (Type I bone loss with menopause or Type 2 bone loss with advanced aging) or for secondary osteoporosis such as glucocorticoid-induced osteoporosis (i.e., steroid treatment exceeding 3 months). In contrast, astronauts are healthy individuals who do not manifest similar deficits in DXA-aBMD after spaceflight but do experience monthly aBMD loss rates that exceed the loss rates of aBMD loss at comparable sites in aged humans [33]. Therefore, while the cellular etiology of osteoporosis may inform therapeutic approaches for bone loss mitigation, there are multiple factors during spaceflight that can contribute to the cellular mediation of spaceflight-induced bone loss which can likewise affect the bone density and structure in distinct fashions. Hence, it is prudent to employ higher-resolution diagnostic modalities, beyond DXA testing for age-induced fragility, to fully understand and evaluate the full influence on skeletal integrity.

As previously alluded, the consequence of spaceflight-related bone loss is likely sub-clinical after space missions of 6 months (postflight aBMD T-scores > −2.5) [9]. Ironically, the inability to detect skeletal fragility `by World Health Organization T-score standards was evident in ISS astronauts before the ARED was flown on-orbit [9] which further highlights the insufficiency of DXA-measured aBMDs as the only standards of astronaut skeletal health, especially after exposure to long-duration spaceflight. Hence, concerns about the maintenance of skeletal health still persist especially during prolonged spaceflight missions (>12 months) or in the event of impaired exercise hardware or an in-flight injury.

While the postflight effect of in-flight interventions on lifetime fracture incidence in long-duration astronauts may be difficult to analyze (i.e., due to small sample sizes, younger age of active astronauts, and self-reporting by retirees), fracture events are the gold standard metric assessing the risk; consequently, the surveillance of fractures in active and retired astronauts is an on-going lifetime process. A recent analysis of fracture data, following a survey of astronauts (n = 262) across 8433.6 person-years, suggested an increased rate of hip and spine fractures in astronauts following spaceflight exposure of 3 mos. or greater (but typically 6-mos) when compared to person-years prior to spaceflight or to no spaceflight at all [50].

As mentioned, characterizing fractures in the astronaut corps has its challenges. With so few astronauts available to study, and even fewer at an age when fractures would be expected to manifest, statistical power may be insufficient to detect differences in fracture risk. Therefore, whether the observed changes in bone structure and mass, that occur during long-duration spaceflight, are predictive of long-term fracture risk may never be substantiated. Increased fracture outcomes in the astronaut cohort may likely not occur soon enough to support the implementation of prophylactic therapies. Therefore, decision makers are faced with evaluating bisphosphonate usage during long-duration missions using knowledge of cellular mechanisms of action and drawing analogies to terrestrial clinical trials. Currently, bisphosphonates are approved for spaceflight use on a case-by-case basis by agreement between the astronaut and his/her flight surgeon. Thus far bisphosphonates have only been used during spaceflight research.

Part of the reasoning for this current practice is due to (i) adverse effects observed with bisphosphonates in terrestrial medicine and (ii) the perception that the risk for adverse side effects outweighs the benefit of taking a pharmaceutical therapy. However, it is the opinion of the authors that bisphosphonate should be used as a preflight preventive for stimulated bone resorption during spaceflights assessed by elevated levels of bone resorption biomarkers assayed in urine specimens collected during flight. The bisphosphonate class of drugs specifically targets osteoclast cells which release the bisphosphonate molecule from the bone matrix during the resorption of the bone matrix. The cellular response of bone to a reduced weight-bearing function has been characterized in iliac crest bone biopsies of subjects skeletally unloaded by 12 weeks of bed rest [34]. In this study, indices of bone resorption (e.g., osteoclast-covered surfaces and eroded surfaces in both trabecular and cortical bone) were increased while the osteoblast-covered surfaces were reduced [34]. With the use of an intravenous bisphosphonate such as Zoledronic Acid, the prevention of bone resorption for the duration of spaceflight-induced bone loss may be induced by a single intravenous administration of Zoledronic Acid given several months prior to spaceflight. This assertion is based upon its reported therapeutic effects which could last as long as the durations of planned missions aboard the ISS, long duration Artemis or Mars missions [51,52]. Adverse side effects have been observed following a single injection of the bisphosphonate, but these effects are usually recognized to be short-term and mild, to occur soon after administration, or can be mitigated with pretreatment with 25-hydroxyvitamin D (with/without oral calcium supplementation) or oral bisphosphonates [53]. The prophylactic infusion of zoledronic acid would allow the management of any acute reactions that are more likely to occur preflight while on Earth [47]. For the astronauts who have already had or have shown a propensity for renal stones, there is an abundance of terrestrial data that validate the ability of bisphosphonate to lower urine calcium and decrease the potential for the development of renal stones suggesting a side benefit for bisphosphonate use [54]. The risk for renal stone formation during spaceflight is heavily based upon increased bone turnover, reduced hydration, and saturated urine in astronauts, although there are additional factors, such as high ambient CO_2_, high salt space diet [55], and ionizing radiation, whose contributions to bone resorption are not fully defined [29,56]. Additionally, we contend that crewmembers who sustain injuries during a space mission—that would limit their ability to exercise at a level to maintain preflight levels of bone mineral content in skeletal bones—would benefit from the anti-resorptive action of bisphosphonates. These individuals will likely not be evacuated back to Earth but will continue on spaceflight missions with reduced physical activity. Astronauts will also be engaged in physical activities on the Martian surface. It is unlikely that the fractional gravity that exists on planetary surfaces (1/3 G on Mars) would fully protect the skeleton from losing bone mass [57]. Humans additionally cannot “feel” weakened bones due to the asymptomatic nature of bone loss. Hence, an astronaut, deconditioned over an extended spaceflight exposure (e.g., a 6-month trip to Mars), would likely be at an increased risk for musculoskeletal injury while performing physical activities on a hazardous planetary surface or if returning to mechanically loaded, preflight activities (e.g., after return to Earth) before skeletal integrity has fully recovered.

Given the challenges of translating spaceflight-induced changes to fracture risk or substantiating fracture consequences of space-related skeletal changes, bisphosphonate therapy—with its ability to suppress the bone-resorbing actions of stimulated osteoclasts—remains an important part of spaceflight preventative health care for astronauts.

## 4. Conclusions

There is evidence suggesting profound degradation occurring in the trabecular bone of astronauts—changes that are unrelated to aging, cannot be resolved by standardized DXA imaging, or prevented by mechanically loading bones by resistive exercise alone. Some of these changes may be irreversible, as suggested by continued loss and delayed recovery in astronauts after return to Earth, or could induce unacceptable outcomes, especially fractures. Further, there is a strong suspicion that, based on analogies to terrestrial disease, these changes could place astronauts at risk for long-term adverse health outcomes. Bisphosphonates have been studied in long-duration astronauts and the data collected from these studies suggests that this class of drug reduces the loss of trabecular bone mineral density and helps preserve total bone mass in space as well as prevent hypercalciuria (and renal stone formation). This class of pharmaceuticals has been used extensively in terrestrial medicine and has a well-documented side effect profile. Additionally, the issue of bone atrophy—specifically in the trabecular compartment and in trabecular microarchitecture–warrants further evaluation using advanced imaging techniques for testing novel therapeutics and countermeasures.

Given all the above information, it is the opinion of the authors that bisphosphonates should be approved as an in-flight countermeasure and considered for use in appropriate long-duration astronauts aboard future missions—missions that may well be even farther and longer than those preceding.

## Figures and Tables

**Figure 1 cells-13-01337-f001:**
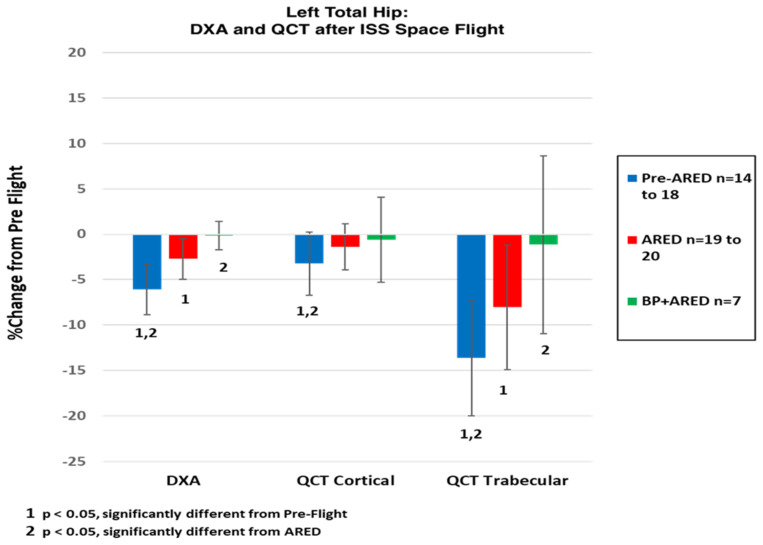
Differential Effects of Spaceflight and of ARED Exercise as Influenced by Bisphosphonate Supplementation. Changes in DXA, QCT-cortical, and QCT-trabecular bone mineral density of the left total hip (from preflight to postflighttesting) were compiled from previously published reports of astronauts from whom informed consents were acquired [24,32,33]. Pre-ARED represented data acquired from crewmembers before ARED was flown on-board the ISS. The access and use of ARED alone (red) were protective against spaceflight losses in QCT-cortical bone but not for changes detected by DXA aBMD and by QCT-measured trabecular bone. Bisphosphonate use combined with ARED (green) was associated with no significant change between pre- and post-flight measurements using DXA and QCT-cortical bone and QCT-trabecular bone.

**Figure 2 cells-13-01337-f002:**
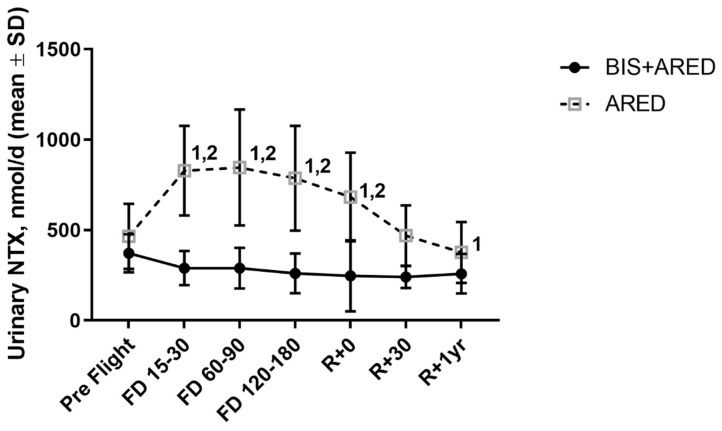
Percent change of level of N-Telopeptide Cross-links (NTX), a marker of bone resorption, from pre-flight levels at increasing flight days (FD). An increase in bone resorptive biomarker NTX is observed when resistive exercise is utilized in isolation (hollow square). When bisphosphonates are used to supplement resistive exercise (solid square), the level of NTX remains stable at pre-flight levels Bis: bisphosphonate; iRED: Interim Resistive. Exercise Device; ARED: Advanced Resistive Exercise Device [35]. Superscript 1 denotes a significant Within-Group delta change in assay results when compared to pre-flight specimens; superscript 2 denotes a significant Between Group delta change between the ARED vs. Bis+ ARED treatments at the specified time point. (Reprinted/adapted with permission from [35]. 2019, Sibonga).

**Figure 3 cells-13-01337-f003:**
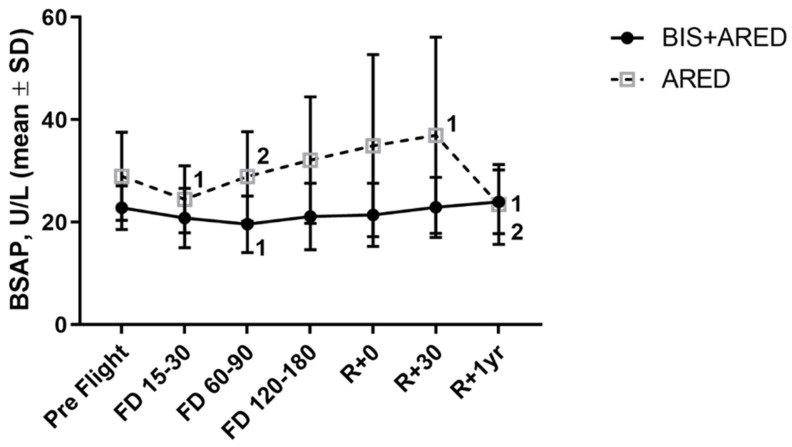
Percent change of level of bone-specific alkaline phosphatase (BSAP), a bone formation biomarker, from pre-flight levels at increasing flight days (FD). ARED alone leads to an increase in BSAP with increasing mission durations (solid circle), while bisphosphonate addition (solid square) tempers this effect perhaps due to a decrease in overall bone turnover. Bis: bisphosphonate; iRED: Interim Resistive Exercise Device; ARED: Advanced Resistive Exercise Device [35]. Superscript 1 denotes a significant Within-Group delta change in assay results when compared to pre-flight specimens; superscript 2 denotes a significant Between-Group delta change between the ARED vs. Bis+ ARED treatments at the specified time point (Reprinted/adapted with permission from [35]. 2019, Sibonga).

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
