# Peer review of "The Case for Bisphosphonate Use in Astronauts Flying Long-Duration Missions"

_cells, 2024, doi:10.3390/cells13161337_

Round 1

Reviewer 1 Report

Comments and Suggestions for Authors

In this review paper manuscript, the authors provided a comprehensive view of space environmental responses in bone tissue in comparison to aging-related changes. It is helpful for the field to provide critical insights on similarities and differences in bone homeostasis in space and aging, with potential issues on the use of currently available drug bisphosphonate. There were no issues found in this manuscript, and publication is supported by this reviewer.

Minor notes:

1. In figures 2, and 3, "1" and "2" labeling were not explained.

2. Many double-spaces were seen in the text.

Reviewer 2 Report

Comments and Suggestions for Authors

In this review, the authors make a case for the use of biphosphonates, drugs that have long been used with success to control osteoclast-mediated bone disease on Earth, to prevent bone loss  in the microgravity of space. In this regard, it is well established that astronauts and cosmonauts lose 1.0% to 1.5% of their bone mass for every month they spend in space and that this loss is due to impairment of osteocyte and osteoblast function and the consequent upregulation of osteoclast-mediated bone resorption. Biphosphonates are incorporated into the bone matrix and are ingested by osteoclasts, causing their apoptosis, thereby  providing a rationale for their use in space travelers. However, not noted by the authors, biphosphonates inhibit the stimulatory activity of osteoclasts on osteoblast differentiation and, as a consequence, persons on these drugs suffer from a blockade of de novo bone formation. Non-the-less the authors provide an excellent review of the effects of microgravity and senescence on skeletal integrity and make a plausible argument for the use of biphosphonates in persons subjected to long duration space flights

Comments on the Quality of English Language

English quality is high.
